# Importance of Adequate qPCR Controls in Infection Control

**DOI:** 10.3390/diagnostics11122373

**Published:** 2021-12-16

**Authors:** Matthew Oughton, Ivan Brukner, Shaun Eintracht, Andreas I. Papadakis, Alan Spatz, Alex Resendes

**Affiliations:** 1Faculty of Medicine, McGill University, Montreal, QC H3G 2M1, Canada; shaun.eintracht@mcgill.ca (S.E.); alan.spatz@mcgill.ca (A.S.); 2Lady Davis Institute for Medical Research, Montreal, QC H3T 1E2, Canada; andreas.papadakis@ladydavis.ca (A.I.P.); alex.resendes@mail.mcgill.ca (A.R.); 3Department of Medicine, Jewish General Hospital, Montreal, QC H3T 1E2, Canada

**Keywords:** infection, control, pandemic, qPCR, sample, quality, assay, clinical, laboratory, quality, false negative

## Abstract

Respiratory screening assays lacking Sample Adequacy Controls (SAC) may result in inadequate sample quality and thus false negative results. The non-adequate samples might represent a significant proportion of the total performed tests, thus resulting in sub-optimal infection control measures with implications that may be critical during pandemic times. The quantitative sample adequacy threshold can be established empirically, measuring the change in the frequency of positive results, as a function of the numerical value of “sample adequacy”. Establishing a quantitative threshold for SAC requires a big number/volume of tests to be analyzed in order to have a statistically valid result. Herein, we are offering for the first time clear clinical evidence that a subset of results, which did not pass minimal sample adequacy criteria, have a significantly lower frequency of positivity compared with the “adequate” samples. Flagging these results and/or re-sampling them is a mitigation strategy, which can dramatically improve infection control measures.

The accuracy of diagnostic test results are dependent on adequate samples [1,2,3,4,5,6,7,8,9]. Current clinical protocols allow for a variety of samples types to be used for the detection of respiratory pathogens, including various anatomical sites and sampling techniques, each having its own respective interpretation of sampling adequacy [1,6,10,11,12,13,14]. Nasopharyngeal swabbing is one of the most common methods for obtaining clinical specimens [4,9,13,15,16,17,18]. However, the human genome equivalents present in the respiratory sampling can vary over one million-fold, while the ratio of virus genome equivalents to human genome equivalents can differ by up to one billion-fold (from (1/3 × 10)^4^ to 3 × 10^4^ ratios) [3,4,19,20]. This inherent variability in both human and virus genome equivalents can be measured with high resolution techniques, like quantitative PCR, by following the quantitative signal of sample-specific biomarkers that must be present in every sample [2,3,4,6,7,8,21,22]. Analysing the presence of this biomarker is performed by incorporating a Sample Adequacy Control (SAC) into the diagnostic assay. SAC not only offers assurance of proper assay processing but also establishes the absence of inhibition of nucleic acid amplification.

To estimate the impact of sampling variability on respiratory swab results, we measured positivity rates (numbers of positive tests/total number of tests) of three common respiratory viruses (influenza A, influenza B, and RSV) as a function of the number of human genome equivalents present in the sample. The concentration of single-copy-human-gene (RNase P) is typically chosen to present the quantitative measure of SAC [3,4,6,7,20,22,23]. In the case of symptomatic respiratory infections, it is usually assumed that the quantity of a virus-specific biomarker is a few log values higher than the sample adequacy biomarker. This can lead to the incorrect conclusion that the virus positivity is “guaranteed”, regardless of sample adequacy. However, by using a larger clinical data set, different scenarios became evident, which explains the importance of routinely determining sampling adequacy. For example, during the viral incubation period, the ratio between viral and human genome equivalents can be very low (~1/10,000) [3], which can differ greatly from what is seen in a patient during a typical symptomatic infection. Thus, in the early stages of viral infections, determination of sample adequacy can be extremely valuable [4,17,18,24,25,26,27] by ensuring the quality and quantity of sampling was sufficient, and greatly reducing the possibility of a false negative result. In addition, consistent use of SAC allows earlier detection of positive cases and improves infectious control measures.

To demonstrate the impact of sampling variability on “missing” positive results, we measured the frequency of positive results as a function of sample adequacy biomarker. The respiratory samples were subjected to microbiology laboratory screening during the period of 2016 to 2018. Group A comprises of 4168 samples that were tested by qPCR on a Roche Light Cycler 480, following a described protocol [15], which excludes nucleic acids extraction. Group B includes 2457 samples that underwent standard nucleic acid isolation (bioMerieux, easyMAG) [3] prior to testing. Samples belonging to the same methodological protocol were divided into two subgroups. These subgroups were defined by the SAC cycle threshold values (Cq) below and above 35 (group A) and 30 Cq units (group B), see Table 1. The number of pathogen positive cases seemed significantly lower when Cq values of SAC were in the range of 35 to 40 Cq units for group A and 30 to 40 Cq units, for group B. To assess the hypothesis that the virus positivity rates are dependent on to the sample adequacy values, the difference in the positivity for each subgroup was analyzed by chi-squared test. The chi-squared test detected a statistically significant decrease in the frequency of positive samples, in both A and B groups independent from the methodology used to perform the assay. The positivity rates decrease 4-fold for group A (chi-square = 92.2, *p*-value < 0.0001) and 2-fold for group B (chi-square: 12.7; *p*-value < 0.0004) as a function of the Cq of SAC.

In general, the frequency of disease-specific biomarker changes must be analysed as a function of sample-specific biomarker changes. For example, in group A, samples having Cq of SAC < 35, have an average virus positivity rate of 0.24 (SD = 0.05). On the contrary, the rest of samples are having Cq of SAC in the range 40 > Cq > 35, but show a significant drop in detecting positives, down to the average of 0.06 (SD = 0.04). A threshold in reporting negatives “with compromised sample adequacy” should be the Cq value of SAC, when the decrease in detecting positives becomes a statistically significant “trend” (*p* < 0.001).

In the ideal sampling case, disease-specific biomarker rate changes should be independent from the sample-specific biomarker changes. Therefore, if there is a significant drop in the disease-specific positivity rate as a function of SAC, the information about sample quality/quantity should be reported. The analyses demonstrate that the negative test result, characterized by the high Cq value of sample adequacy biomarker, might benefit from resulting as “inadequate sample quality and/or quantity”, and suggesting repeat sampling.

Although study results are focused on pre-pandemic common respiratory viruses as a function of sample adequacy biomarker, they may have important implications in the context of SARS-CoV2. With the positivity rate of respiratory infection in some jurisdictions as high as 30–40% [28,29,30], inadequate samples (10–15%) may produce 50–75% of false negative results on a daily basis. An assay which does not contain sample adequacy measures lacks the capacity to correct sampling errors. It also impairs early detection of disease and consequent public health efforts to prevent community transmission. Sample Adequacy Control remains still uncommon in individual commercial assays, intended for the fast and accurate testing of COVID-19 and other respiratory pathogens, despite explicit recommendations made by the Word Health Organisation and Centers for Disease Control and Prevention (USA). However, this trend is changing. Some examples of multiplex respiratory panels of Luminex [31] and BD Max [32], together with cartridge-based qPCR test Idylla™ SARS-CoV-2/Flu/RSV from Biocartis [33], or fast isothermal kits of Lucira [34] and Cue [35], are updating quality features of tests, by including RNAse P as a control for sample adequacy. The technical description of how to incorporate SAC into Nucleic Acid Amplification (NAA) assays is described in multiple prior publications [3,4,5,6,7,8,14,17,20,21,23].

## Figures and Tables

**Table 1 diagnostics-11-02373-t001:** Relative frequency of positive results is affected by the Cq values of sample adequacy controls.

Number of tested samples	4234		2538	
Sample processing methodology	Group A (direct qPCR)		Group B (isolated nucleic acids)	
SAC range based on Cq	<35	35 < Cq < 40	<30	30 < Cq < 40
Relative positivity	1	0.25	1	0.5
Significance of Chi-Square	*p* < 0.0001		*p* < 0.0004	

**Legend: Samples were tested either through direct qPCR (Group A) or qPCR after standard nucleic acid (N.A.) isolation (Group B)**. Samples having Cq value of SAC lower then 35 and 30 Cq units, for group A and B respectively, are characterised by the arbitrary positivity of one. The drop from this value is 4-fold for group A and 2-fold for group B, both characterised as statistically significant; Chi-Square p-values (0.0001 and 0.0004, respectively).

## Data Availability

Not applicable.

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
