# Peer review of "Importance of Adequate qPCR Controls in Infection Control"

_diagnostics, 2021, doi:10.3390/diagnostics11122373_

Round 1

Reviewer 1 Report

Reference number: MS 1498576

Title: Importance of adequate qPCR controls in infection control

Corresponding Authors: Drs Matthew Oughton and Ivan Brukner

Contributing Authors: Shaun Eintracht, Andreas I. Papadakis, Alan Spatz and Alex Resendes

A brief summary: The communication is offering an approach how to improve the accuracy of diagnostic test for detection of respiratory pathogens. The application of proposed approach could ensure more reliable results of pathogens detection in the sample.

General concept comments: Considering that nasopharyngeal swabbing is used as clinical specimens it is of great importance to distinguish the origin of DNA in the sample. Therefore, a proper control must be included in the qPCR testing. Using of single-copy-human-gene like RNase P is an adequate marker to improve Sample Adequacy Controls (SAC) to avoid false-negative results as much as possible. Presented results are well documented. Relevant references are presented helping reader to get more information regarding the topic of communication.

Specific comments:

Line 76: It is written: “Legend of Table 1”.  – It should be changed into “Legend:” since it is given under Table 1.

Author Response

Review 1

Reference number: MS 1498576

Title: Importance of adequate qPCR controls in infection control

Corresponding Authors: Drs Matthew Oughton and Ivan Brukner

Contributing Authors: Shaun Eintracht, Andreas I. Papadakis, Alan Spatz and Alex Resendes

A brief summary: The communication is offering an approach how to improve the accuracy of diagnostic test for detection of respiratory pathogens. The application of proposed approach could ensure more reliable results of pathogens detection in the sample.

General concept comments: Considering that nasopharyngeal swabbing is used as clinical specimens it is of great importance to distinguish the origin of DNA in the sample. Therefore, a proper control must be included in the qPCR testing. Using of single-copy-human-gene like RNase P is an adequate marker to improve Sample Adequacy Controls (SAC) to avoid false-negative results as much as possible. Presented results are well documented. Relevant references are presented helping reader to get more information regarding the topic of communication.

Specific comments:

Line 76: It is written: “Legend of Table 1”.  – It should be changed into “Legend:” since it is given under Table 1.

The specific comment at Line 76 is corrected. Thank you for encouragements and recognition of importance of this work for the future of qPCR testing of infectious diseases.  We appreciate the time and the effort of reviewer.

Reviewer 2 Report

This report presents a case study on the impact of variations in sample quality on diagnostic tests using data from large-scale respiratory sample testing. Based on a commonly used measure for sample quality, SAC, the report illustrates significant variation in test positivity rates indicating the need for accounting for SAC when evaluating the quality of a diagnostic test. 

Overall, this is a report of interest given the ongoing COVID-19 pandemic where false negative results can significantly impact public health outcomes. In my opinion, the manuscript can be strengthened in some key areas to be appropriate for publication in Diagnostics.

The choice of the SAC thresholds for the two groups of samples seem arbitrary. The authors can illustrate through appropriate figures how the test positivity rates vary with different choices of SAC thresholds for each of these experiments, and motivate a more data-driven approach to pick these thresholds for individual experiments.

The phrasing of several statements in the manuscript can be sharpened: 

“fundamentally requires adequate samples”.

“a diverse variety of samples”.

These statements are too vague at summarizing a large number of references

Please do a thorough proofreading to correct typographical and syntax errors such as:

Page 2. “are in relation”

“independent on”

Author Response

Review 2

This report presents a case study on the impact of variations in sample quality on diagnostic tests using data from large-scale respiratory sample testing. Based on a commonly used measure for sample quality, SAC, the report illustrates significant variation in test positivity rates indicating the need for accounting for SAC when evaluating the quality of a diagnostic test.

Overall, this is a report of interest given the ongoing COVID-19 pandemic where false negative results can significantly impact public health outcomes. In my opinion, the manuscript can be strengthened in some key areas to be appropriate for publication in Diagnostics.

The choice of the SAC thresholds for the two groups of samples seem arbitrary. The authors can illustrate through appropriate figures how the test positivity rates vary with different choices of SAC thresholds for each of these experiments, and motivate a more data-driven approach to pick these thresholds for individual experiments.

We want to thank referee #2 for suggestions and comments that strengthen the Letter. We’ve added text, modified content and information, and corrected errors noticed to address comments pointed out by this referee. We agree that “data-driven approach is essential to pick these thresholds for individual experiments.” We added the following sentence: 

“In general, the frequency of disease-specific biomarker changes must be analysed as a function of sample-specific biomarker changes.  For example, in group A, samples having Cq of SAC <35, have an average virus positivity rate of 0.24 (SD =0.05). On the contrary, the rest of samples are having Cq of SAC in the range 40>Cq>35, but show a significant drop in detecting positives, down to the average of 0.06 (SD=0.04).  A threshold in reporting negatives “with compromised sample adequacy” should be the Cq value of SAC, when decrease in detecting positives becomes statistically significant “trend” (P<0.001).

In the ideal sampling case, disease-specific biomarker rate changes should be independent from the sample-specific biomarker changes. Therefore, if there is a significant drop in the disease-specific positivity rate as a function of SAC, the information about sample quality/quantity should be reported. The analyses demonstrate that the negative test result, characterized by the high Cq value of sample adequacy biomarker, might benefit from resulting as “inadequate sample quality and/or quantity”, and suggesting repeat sampling.”

We include the following table for referee #2, which illustrate the drop in positivity rate), explaining simplicity of operations. The red characters are samples having SAC Cq >35.  We will omit this information/data in the final version of MS, due to the need for simplicity and the short message (communication).

direct qPCR

Average +/- SD

Cq value of Rnase P

number of samples

positives for FluA/B/RSV

 positivity rate

positivity rate for SAC interval

23

5

1

0.2

0.248+/-0.05

24

20

4

0.2

25

44

7

0.159090909

26

95

25

0.263157895

27

167

49

0.293413174

28

249

75

0.301204819

29

347

98

0.282420749

30

459

148

0.322440087

31

577

164

0.284228769

32

634

158

0.249211356

33

646

141

0.218266254

34

431

89

0.20649652

35

260

15

0.057692308

0.06+/-0.04

36

131

9

0.06870229

37

60

4

0.066666667

38

24

3

0.125

39

11

1

0.090909091

40

7

0

0

The phrasing of several statements in the manuscript can be sharpened:

“fundamentally requires adequate samples”.

Thank you, Replaced by:

The accuracy of diagnostic test results is dependent on samples quality and quantity.

“a diverse variety of samples”.

“Diverse” is removed and replaced with

“Current clinical protocols allow for a variety of samples to be used for the detection of respiratory pathogens, including various anatomical sites, tissues, sample types and sampling techniques, each having its own respective interpretation of sampling adequacy”

Please do a thorough proofreading to correct typographical and syntax errors such as:

Page 2. “are in relation”

Replaced with

“To assess the hypothesis that the virus positivity rates are dependent on to the sample adequacy values, the difference in the positivity for each subgroup was analyzed by Chi-Square Test.”

“independent on

replaced with “independent from”